# Possible Crosstalk and Alterations in Gut Bacteriome and Virome in HIV-1 Infection and the Associated Comorbidities Related to Metabolic Disorder

**DOI:** 10.3390/v17070990

**Published:** 2025-07-16

**Authors:** Komal Shrivastav, Hesham Nasser, Terumasa Ikeda, Vijay Nema

**Affiliations:** 1Division of Molecular Biology, ICMR-National Institute of Translational Virology and AIDS Research (Formerly NARI), Pune 411026, India; kshrivastava880@gmail.com; 2Division of Molecular Virology & Genetics, Joint Research Center for Human Retrovirus Infection, Kumamoto University, Kumamoto 8600811, Japan; hesham-n@kumamoto-u.ac.jp (H.N.); ikedat@kumamoto-u.ac.jp (T.I.); 3Faculty of Biological Sciences, Academy of Scientific and Innovative Research (AcSIR), Ghaziabad 201002, India; 4ICMR-National Institute of Research in Tribal Health, ICMR-NIRTH Campus, Jabalpur 482003, India

**Keywords:** human immunodeficiency virus-1, metabolic disorder, gut microbiome, virome, type-2 diabetes mellitus, cardiovascular diseases, crosstalk

## Abstract

Improved antiretroviral therapy (ART) has significantly increased the life expectancy of people living with HIV (PLWH). At the same time, other complications like metabolic syndrome (MetS) are coming up as new challenges to handle. This review aims to explore the emerging evidence of gut microbiome and virome alterations in human immunodeficiency virus-1 (HIV-1) infection and associated metabolic disorders, such as type-2 diabetes (T2DM) and cardiovascular disease (CVD), with a focus on their interplay, contribution to immune dysfunction, and potential as therapeutic targets. We conducted a comprehensive review of the current literature on gut bacteriome and virome changes in HIV-1-infected individuals and those with metabolic comorbidities emphasizing their complex interplay and potential as biomarkers or therapeutic targets. HIV-1 infection disrupts gut microbial homeostasis, promoting bacterial translocation, systemic inflammation, and metabolic dysregulation. Similarly, metabolic disorders are marked by reduced beneficial short-chain fatty acid-producing bacteria and an increase in pro-inflammatory taxa. Alterations in the gut virome, particularly involving bacteriophages, may exacerbate bacterial dysbiosis and immune dysfunction. Conversely, some viral populations have been associated with immune restoration post-ART. These findings point toward a dynamic and bidirectional relationship between the gut virome, bacteriome, and host immunity. Targeted interventions such as microbiome modulation and fecal virome transplantation (FVT) offer promising avenues for restoring gut homeostasis and improving long-term outcomes in PLWH.

## 1. Introduction

The microbial communities residing in the human gut include bacteria, viruses, fungi and parasites that play various roles in keeping the human gut in balance, while its disruption—called dysbiosis—causes different harmful to injurious impacts on the host’s health [1,2]. The gut virome, a microbiome component, particularly bacterial phages, significantly influences human gut microbial ecology, but their role is somewhat less characterized as compared to the various bacterial communities. There are other factors from the host side as well as from other microscopic entities that either regulate the gut environment directly or through the activity of these bacterial entities. A longitudinal study followed by a series of fecal samples for 6 months from a 2-year-old boy having atopic eczema (dermatitis) is an example wherein 31 viral strains were identified. Among this, a lytic crAssphage was found to influence its bacterial host by downregulating aromatic amino acid (AAA) metabolism, thus modulating the immunologically active AAA derivatives. These metabolic changes were also related to the remission of symptoms, depicting the interplay between virome dynamics, microbial metabolism, and human host health outcomes [3]. While earlier studies implicated genetic factors like mutations in the *filaggrin* (*FLG*) gene as major contributors to the pathogenesis of atopic dermatitis (AD) [4,5], the disease is now understood to be multifactorial, and thus the role of microbiota, both skin and gut, is becoming evident in its pathogenesis.

The growth and maturity of the host’s immune system and intestinal epithelium depend on the gut microbiota. This microbiota influences mucus layer characteristics, stimulates lymphoid structure formation, regulates the activation and differentiation of several lymphocyte populations, and maintains equilibrium in the synthesis of immunoglobulin A and antimicrobial peptides. By increasing nutrient sources, generating vital vitamins, and facilitating xenobiotic metabolism, the gut microbiota aids in host metabolism and adiposity (Figure 1). It also influences a variety of other host physiological factors, such as organ morphogenesis, intestinal vascularization, tissue homeostasis, carcinogenesis, bone mass, and behavior [6].

The schematic figure shows how HIV-1 infection disrupts normal gut microbiota, leading to reduced diversity, increased inflammation, and compromised gut barrier function. This dysbiosis contributes to chronic inflammation and the development of metabolic syndrome. Therapeutic strategies aim to restore microbial balance, improve gut integrity, and reduce HIV-1-related metabolic complications.

The role of the gut microbiome is becoming increasingly known in metabolic diseases but its role in infectious diseases is now taking significant orientation.

Beyond AD, the gut microbiome plays a central role in immune homeostasis, and its dysregulation has been implicated in various diseases, including human immunodeficiency virus-1 (HIV-1) infection. Gut-associated lymphoid tissue (GALT) is key to human and simian immunodeficiency virus (HIV and SIV) infection. HIV and SIV target the gut at various levels, including immunological, structural, and microbiological [7,8].

HIV-1 viral replication occurring within CD4+ T lymphocytes, the primary target cells of virus, essentially contribute to the disease pathogenesis; however, HIV-1-induced chronic immune activation is shown to drive the overall disease progression [9]. It is now believed that the gut microbiome plays a central role in HIV-1 immunopathogenesis and associated chronic complications [10,11]. GALT is infected in the early stages of HIV-1 infection, which disrupts gut integrity due to the massive depletion of CD4+ T lymphocytes, including T helper (Th) 17 and Th22 cells. Furthermore, HIV-1 induces enteropathy, mucosal inflammation, and intestinal epithelial cell damage. Indeed, such a damaged intestinal epithelial barrier is a major contributor of HIV-1-associated local and systemic inflammation [12,13] (Figure 1). At the cellular level, the structural damage of the gut barrier is a result of massive enterocyte apoptosis, decreased expression of tight junction proteins, and increased intestinal permeability [14]. Collectively, these abnormalities result in focal breaches of the gut barrier and subsequent microbial translocation at both local and systemic levels [15,16].

Microbial translocation regulates inflammation within the gut, which subsequently initiates intestinal barrier dysfunction (Figure 1). It was demonstrated that transcripts encoding inflammatory cytokines [such as tumor necrosis factor (TNF), interleukin 6 (IL-6), IL-10, and interferon-gamma (IFN- γ)] are significantly upregulated in the colonic mucosa of HIV-1-infected individuals and persist in high levels despite antiretroviral therapy (ART) [17,18]. These inflammatory markers also serve as predictors of disease progression in untreated HIV-1-infected individuals [19].

Of note, metagenomic analysis of people with gut inflammatory diseases has shown reduction in the microbial richness with low gene counts (LGCs) and altered intestinal microbiome (dysbiosis) [20,21,22]. HIV-1 has been widely associated with an increased susceptibility to tuberculosis and a heightened risk of developing metabolic disorders. This review focuses on HIV-1 infection due to its higher hazard risk and significantly greater global prevalence compared to HIV-2 [23,24,25]. Accordingly, the relationship between intestinal microbiome and HIV-1-associated gut mucosal pathogenesis is a two-way street, as HIV-1-associated changes in gut mucosa lead to dysbiosis [26,27,28] and the dysbiosis subsequently disrupts intestinal homeostasis, further contributing to sustained HIV-1-associated immune activation and inflammation (Figure 1) [13,29]. Indeed, an emphasis of the significance of the intestinal microbiome has been concluded by studies focusing on rebalancing intestinal microbiome and mucosal barrier restoration in patients with (Acquired Immuno-deficiency Syndrome) AIDS to improve treatment outcomes [30,31]. Therefore, we aim to explore the interaction among HIV-1 infection, intestinal microbiome dysbiosis, and mucosal barrier damage in HIV-1-infected patients, which is necessary for introduction of treatment strategy to balance the intestinal microecology and improve mucosal immunity, thereby supporting the overall resolution.

As virome–bacteriome interactions may have significant implications for gut health and disease susceptibility, we hypothesize that shifts in gut virome composition contribute directly to the chronic inflammation observed in HIV-1 metabolic comorbidities through phage–host immune interactions [32].

## 2. Search Strategy and Selection Criteria

This review was conducted based on a comprehensive literature search focusing on gut microbiome–virome interactions in HIV-1 infection and associated metabolic disorders, including type-2 diabetes mellitus (T2DM) and cardiovascular disease (CVD). The search was performed using scientific databases such as PubMed, Scopus, and Google Scholar. A combination of Medical Subject Headings (MeSH) terms and keywords were used, including “Gut microbiome in HIV”, “HIV and gut virome”, “phage therapy in metabolic disorders”, “microbial translocation in HIV”, “gut dysbiosis in cardiovascular disease”, “gut dysbiosis in type 2 diabetes mellitus”, “intestinal microbiota alterations in HIV”, “eukaryotic viruses and metabolic disorders”, and “bacteriophage-host interactions in gut health.” Manuscripts that were from the last 5 years and assessed incident clinical events, or included meta-analyses comprehensively summarizing existing data were preferred. Relevant articles were screened by title and abstract, and full texts of selected studies were reviewed for final inclusion. Reference lists of key papers were also examined to identify the additional relevant literature. Only papers published in English were reviewed.

## 3. Diversity of the Components of Gut Microbiome

Before proceeding to a general microbiome profile of the human gut, it is essential to consider the variety and diversity in human geography, lifestyle, diet, age, sex, drug/antibiotic regimens, and stressful conditions [33] that affect the gut microbiota. Thus, the microbiome also significantly varies from individual to individual [1].

### 3.1. Communities Comprising Gut Microbiome

In general, a healthy gut bacteriome comprises predominant short-chain fatty acid (SCFA)-producing bacteria such as Ruminococcus obeum, Bifidobacterium longum, Roseburia intestinalis, Roseburia inulinivorans, Coprococcus comes, Akkermansia muciniphila, Eubacterium rectale, Bifidobacterium adolescentis, and Roseburia hominis. Fecalibacterium prausnitzii is a marker of gut health. SCFA has important functions such as maintaining gut integrity and modulating immunity, as discussed previously. Thus, an abundance of beneficial bacteria that contribute to metabolic health, low inflammation, and immune stability is evident in healthy gut microbiota. There is a balanced ratio of Firmicutes and Bacteroides (SCFA producers) with a low abundance of pathogenic species and opportunistic microbes. Proteobacteria levels are regulated in a healthy gut, preventing dysbiosis [34].

Consistently, the gut virome is a heterogeneous composition of DNA and RNA viruses, including single-stranded (ss) and double-stranded (ds) forms. Phages occupy a significant proportion and dominate with about >90%, whereas eukaryotic viruses account for <10% [35,36]. Predominantly, DNA-genome phages like *Caudovirecetes* dsDNA and *Microviridae* ssDNA are found in the gut. CrAssphage from the *Intestiviridae* family is also a prevalent and stable colonizer in the gut, abundantly found in human feces. A few viral families show higher prevalence, such as *Circoviridae* and *Myoviridae*, which also exhibit greater genus-level diversity. Additionally, an increased abundance of Podoviruses is observed. In one study, the intestinal luminal content of both pigs and macaques showed increased *Caudoviricetes*, including crAss-like phages and ssDNA *Microviridae*, with a lower fraction of eukaryotic viruses (families *Circoviridae*, *Astroviridae*, *Caliciviridae*, and *Parvoviridae* [36].

Moreover, eukaryotic phages in the healthy gut are supposed to be less pathogenic, and even if such pathogenic groups are present, their activity is controlled by healthy immune conditions. They are latent without causing any general harm to the human host. Dormant DNA viral communities of herpesviruses, anelloviruses [37], and adenoviruses are mostly found in the human gut, while RNA viruses are rare and often of plant origin. While commenting on the composition, one thing to note is the instability of RNA over DNA and possible bias towards analysis techniques to identify the RNA genetic material. Also, the gut bacteriome maintains homeostasis with the virome [38].

The phages modulate bacterial population, and the phage–bacteria dynamics are central to the balance of the gut ecosystem, preventing dysbiosis. Stable colonization of temperate phages is associated with a healthy gut with limited lytic activity [36,39].

### 3.2. Impact of Diet on Microbiome Communities

Since the introduction of different environmental cues for infants starts with diet as they age, the viewpoint of how diet impacts our microbiome, including bacteriome, virome, and phageome, is essential [40].

Different regions have various dietary patterns: Western diets are rich in processed foods and high-fat content, while the Indian diet includes more spices and medicinal plants with anti-cancer properties. A vegan diet eliminates all animal-derived protein sources, while a non-vegetarian diet will ensure more protein-rich content; even eating frequency and appetite are different with respect to different individuals. Likewise, the dietary differences in infants viz. formula-fed or breastfed result in distinct gut phageome and bacteriome profiles (Figure 2). 

The schematic diagram illustrates how dietary factors influence gut microbiome. Appetite regulates the food intake and calorie intake governs the chemical energy available for metabolism, producing ATP, heat, carbon dioxide, and water. Diverse dietary patterns including vegan, plant-based, mixed, Western, high-fat, and fiber-rich diets affect the shaping of different microbial communities. Studies showed enrichment of *Microviridae*, *Phycodnaviridae*, and *Mimiviridae* in mice fed a high-fat diet. Dietary fibers promote SCFA production by gut microbes, contributing to host metabolism. Early-life feeding practices like mother or formula feeding also modulate microbial colonization and diversity.

Specific food components, such as wheat and oats, promote beneficial bacteria like *Bifidobacterium* species. Rice suppresses *Bifidobacterium*, *Lactobacillus*, *Ruminococcus*, and *Bacteroides* [41]. The glucose homeostasis of T2DM patients was improved when a high-fiber diet was adopted. They also showed increased abundances of *Lactobacillus*, *Bifidobacterium*, and *Akkermansia*, while *Desulfovibrio*, *Klebsiella*, and other opportunistic pathogens were decreased [42].

A longitudinal study that evaluated viral sequence abundance revealed that gut virome is highly individual-specific, unique, and temporally stable unless exposed to different cues, as a response to which it is also altered [43]. Specific dietary changes, including in fructose and SCFA, can activate prophages from *Lactobacillus reuteri*, thus triggering temperate phages to undergo lytic conversion, which justifies diet as one of the environmental cues, as mentioned previously. A high-fructose diet increases phage production, possibly through the activated cdc42-associated kinase (ACK) signaling pathway. One study reported spatial compartmentalization between viruses present in mucosa and lumen of the gut, with both showing higher *Caudoviricetes* class members. High-fat diet (HFD)-induced mice show an increase in the virus families *Microviridae*, *Phycodnaviridae*, and *Mimiviridae* in the fecal virome (Figure 2). The study reported reduced lysogenic phages in the *Siphoviridae* family [44,45,46].

## 4. Feed-Forward Cycle Between Gut Inflammation and Gut Dysbiosis in HIV-1

HIV-1 can replicate in GALT in early stage of infection and further mediate pronounced changes to intestinal barrier and its local immunity [47,48,49,50]. In GALT, HIV-1 infection induces intestinal microbiota dysbiosis, which eventually makes way for worsening clinical symptoms and could promote the occurrence of comorbidities associated with metabolic disorders [24,25,51]. The keys to these outcomes are a damaged intestinal mucosal barrier and persistent inflammatory response [52].

### 4.1. Effect of Multilevel Gut Homeostasis Disruption in HIV-1 Infection on Intestinal Barrier Integrity

Upon HIV-1 infection, the clinical condition begins when there is rapid and mass depletion of CD4+ T cells as they serve as primary target cells of HIV-1 due to the availability of C-C chemokine receptor type 5 (CCR5) co-receptors. The critical impact of HIV-1 on gut immunity is evident in GALT of humanized mice, too, as HIV-1 led to the depletion of immune cells by infecting nearby cells through cell-to-cell contact regions called virological synapses [53].

The gut barrier is implicated in HIV-1 infection at the structural or anatomical level as well, as it leads to a phenomenon of ‘leaky gut’, which allows entry of microbial antigens and metabolites into the bloodstream. This induces the development of a sustained pro-inflammatory state, which further transforms into a chronic inflammation in treated HIV-1 patients as well. Immune reconstitution inflammatory syndrome (IRIS) in HIV-1 infected individual was detected even after the ART introduction and achieving undetectable viral load, overall immune recovery remained incomplete, leaving the individual at higher risk of further complications [54]. “Viro-immunological discordants” are a group of individuals who fail to recover their CD4+ T cell count despite achieving viral suppression through ART [55]. This might be related to other levels at which HIV-1 impacts the host, i.e., microbiological.

At the microbiological level, the gut bacteriome and virome profiles of a healthy individual differ distinctly from those of an HIV-1-infected individual. HIV-1-infected patients show a general increase in bacterial species associated with higher levels of pro-inflammatory cytokines. So, classes like *Negativicutes*, *Bacilli*, and *Coriobacteria* are increased in abundance, whereas there is depletion of bacterial species associated with anti-inflammatory cytokines like *Clostridia* [51]. A general reduction in SCFA-producing bacteria is seen as accompanied by an increase in species associated with an increase in inflammatory responses [56].

Similar evidence was provided from a study that investigated why some women living with HIV (WLHIV) continue to shed HIV-1 in their genital tract despite effective viral suppression by ART. It was found that women having more shedding of HIV-1 had greater fluctuations in their bacterial microbiome over time, suggesting an unstable microbial environment in their genital tract. Moreover, the virome profile also changes in different conditions. A relation to virome was also seen, for example, in women who did not shed HIV-1, the virome composition, particularly that of anelloviruses, showed changes over time, which could be due to a long-term ART regimen [57].

Anelloviruses are small, single-stranded DNA viruses commonly found in the human gut. They are not known to cause any disease yet, but their alteration is seen with diseased and healthy conditions. Advanced HIV-1 infection is associated with elevated levels of human *Anelloviridae* sequences in the gut virome, especially in individuals with severe immunodeficiency (CD4+ T cells < 200 cells/µL). Longitudinal data show that short-term ART (24 months) significantly reduces *Anelloviridae* abundance. Notably, baseline detection of anellovirus sequences independently predicts poor immune recovery during ART, suggesting potential use as a prognostic biomarker. However, whether this virome shift is causal or merely a correlate of immune dysfunction remains to be fully elucidated [58]. Bacteriophages comprise a vast proportion of the gut virome, and their difference in microbiome composition (beta diversity) shows altered composition in HIV-1 infected patients compared to healthy controls [36]. An increase in phages belonging to the *Caudoviricetes* class (dsDNA) and a decrease in *Malgrandaviricetes* in the treatment-naive group were reported. *Caudoviricetes* are also increased in inflammatory bowel disorder (IBD), indicating the possible link to increased inflammatory state and gut permeability [59].

### 4.2. Multilevel Disruption of Gut Homeostasis in HIV-1 Infection on the Microbial Translocation

Indeed, the translocated microbial products from damaged/inflamed gut could directly stimulate the immune system, and there could be direct or indirect regulation by viruses. ART could induce partial restoration of the altered gut microbiome, but an exact reversion to a healthy state is not observed as evident from decreased alpha diversity in PLWH on long-term ART. In particular, Enterobacteriaceae increases, which is associated with inflammation. *Aeromonas* and *Prevotella*, members of Enterobacteriaceae, are Gram-negative entities that have lipopolysaccharide (LPS) in their cell walls [60].

Lipopolysaccharide (LPS) triggers inflammatory Th1 responses, inhibiting the differentiation of Th17 cells; thus, IL-17 levels are reduced. IL-17 proves to be essential for the expression of tight junction proteins claudin-1 and claudin-2. Its reduction worsens gut integrity, promoting a leaky gut. It also induces a depletion of CD4+ T cells [14].

*Prevotella* is seen to be increased in HIV-1 infection, although this observation is also seen in men who have sex with men (MSM), irrespective of their HIV-1 status. *Negativicutes*, *Bacilli*, and *Coriobacteriia* are also Gram-negative bacteria that have an outer membrane with LPS. They are positively correlated with IFN-γ and IL-1β plasma levels. Bacterial families like Erysipelotrichaceae within the class *Bacilli* and Atopobiaceae within *Coriobacteriia* are more abundant in the gut microbiota of HIV-1 infected patients. These are negatively correlated with anti-inflammatory cytokines IL-19 and IL-35. Additionally, the genus *Prevotella* is also enriched in PLWH [61].

The enrichment of *Prevotella* and the decrease in anti-inflammatory cytokines further emphasize the role of gut microbiota in influencing health outcomes for PLWH [51]. The enriched bacterial species in HIV-1 infection induce chemokines—like monocyte chemoattractant protein 1/C-C motif chemokine ligand 2 (MCP-1/CCL2), MCP-4/CCL13, and macrophage inflammatory protein 1α (MIP-1α/CCL3)—involved in the recruitment of monocytes to inflammation site, thus exacerbating the inflammation [51]. A leaky gut makes microbial translocation easier and more convenient to the bloodstream and amplifies systemic inflammation, thus creating a vicious cycle of dysbiosis and immune activation in HIV-1 infection. At the same time, lower pathogenicity and slower disease progression of HIV-2 can be linked with preserved intestinal epithelial integrity [51,62]. *Bacteroides* are also Gram-negative bacteria, but they are essential SCFA producers along with *Lachnospiraceae* and *Ruminococcaceae*. SCFAs are beneficial for many reasons: butyrate is the primary energy source for gut epithelial cells. It promotes the expression of proteins forming tight junctions like occludin, claudins, and zonula occludens-1 (ZO-1). SCFA tends to suppress the production of pro-inflammatory cytokines and subsequently increase the production of anti-inflammatory cytokines by modulating signaling pathways. They are shown to stimulate mucus production by goblet cells and lower the luminal pH so that the acidic pH will inhibit the growth of pathogenic bacteria; finally, they also have a role in immune modulation as they can interact with G-protein-coupled receptors (GPCRs) [63]. Since the discussion involves orders and families, different genera belonging to the same family or order may show different overall impacts. Hence, it is bound to show contrasting data availability from the same group of organisms, as observed in a few studies wherein the depletion of *Bacteroides* members was associated with markers of immune disruption, T cell activation, and chronic inflammation in HIV-1-infected individuals. The extent of bacterial dysbiosis generally correlates with the activity of the kynurenine pathway of tryptophan catabolism and plasma concentration of IL-6 [51].

The complex interplay of host metabolites, gut commensal bacterial communities, and inflammation in the host in the context of HIV-1 infection has been examined. SIV has been widely recognized as a model for HIV research, with pathogenic SIV infection being associated with a change in the enteric virome, as 32 enteric viruses were identified during infection in one study. It reported adenovirus infection associated with enteritis and parvovirus viremia in animals with advanced AIDS, thus giving proof that viruses also show dysbiosis in SIV infection [64].

For the logical extrapolation of this data to humans, the vast majority of phages and eukaryotic viruses must be considered, as they outnumber the gut bacterial count [38,44]. So, the regulatory role of phages on bacteriome is vital to counteract and mitigate the gut dysbiosis discussed earlier. Viruses tend to colonize early after birth in the human gut. A study followed virome analysis of infant fecal samples from birth to age one and reported that most of the viral reads were of six families of bacteriophages, including five dsDNA virus families of the class *Caudoviricetes*, with *Siphoviridae* and *Podoviridae* being the most abundant. Thirty-four eukaryotic virus families were found wherein animal viruses *Anelloviridae*, *Astroviridae*, *Caliciviridae*, *Genomoviridae*, *Parvoviridae*, *Picornaviridae*, and *Reoviridae* comprised 98% reads with one plant virus—*Virgaviridae* [65]. Another study reported that higher HIV-1 replication leads to an abundance of anellovirus, which correlates negatively with CD4+ T cell counts, although it’s important to note that short-term ART of 24 months was initially linked to a significant decrease in human *Anelloviridae* sequences. PLWH with severe immunodeficiency (CD4+ T cells < 200 cells/µL) has an abundance of *Anelloviridae*, *Adenoviridae*, and *Papillomaviridae* [58]. Pegivirus, on the other hand, is increased with CD4+ T cell counts and decreases as the HIV-1 viral load increases, thus indicating that gut eukaryotic viruses have different dynamics during infections [66].

One thing to note is that *Anelloviridae* and Pegivirus are DNA and RNA viruses, respectively. Thus, ART could have acted on Pegivirus directly, leading to such effects and having some other mechanism of action to deal with anellovirus. Hence, eukaryotic viruses could use direct and indirect approaches to interact with human hosts [66]. This crosstalk is possible directly or indirectly mediated via metabolites. Another link of the *Anelloviridae* family, particularly the Torque teno virus (TTV), was found in colorectal cancer (CRC) as they were more abundant in CRC tissues than in healthy controls. Also, higher levels of TTV were found in intestinal lamina propria; thus, it can be inferred that anelloviruses were consistently associated with diseased or inflammatory conditions [67]. Bacteriophages also show dysbiosis, as seen by decreased *Inoviridae* sequences in stool samples of PLWH with CD4+ T cells < 200 cells/µL [68]. TTV microRNAs (miRNAs) downregulate N-MYC interactor (NMI) proteins associated with the IFN pathway while open reading frame 2 (ORF2) protein inhibits nuclear factor kappa B (NF-κB) translocation into the nucleus, reducing the inflammatory mediator’s [IL-6, IL-8, and Cyclo-oxygenase-2 (COX-2)] expression. TTV genogroup 4 with cytosine–phosphate–guanine (CpG) motif [69] can activate Toll-like receptor 9 (TLR9), stimulating pro-inflammatory cytokine IFN-γ, IL-6, and IL-12 release. Thus, TTVs may exhibit both stimulatory and inhibitory CpG motifs, implicating their role in balancing inflammation. Dysregulated TTV activity may contribute to chronic immune activation, a hallmark of HIV-1 disease progression [70,71,72].

Additionally, variations in TTV miRNA profiles between HIV-1-positive individuals and healthy controls suggest their involvement in metabolic disorders and gut inflammation [37,69]. The increased prevalence of eukaryotic viruses in PLWH highlights their potential role in exacerbating gut dysbiosis and inflammation, promoting a feed-forward cycle of immune dysfunction and microbial imbalance. This aligns with evidence of viral expansion in stool samples from HIV-1-positive individuals, underscoring the need to explore the interplay between eukaryotic viruses and other gut microbiome components in shaping disease outcomes [52].

### 4.3. Intestinal Dysbiosis Contributes to HIV-1-Associated Metabolic Changes

Notably, chronic inflammation, immune activation, and ART-associated metabolic changes are key factors predisposing PLWH to comorbidities [73]. HIV-1-associated comorbidities related to metabolic disease primarily include CVD and diabetes, particularly T2DM [74]. A study on the gut dsDNA virome in school-aged children with normal weight (NW), obesity (O), and obesity with metabolic syndrome (OMS) revealed that phage richness and diversity increased in O and OMS. At the same time, the abundance of virus-like particles (VLPs) remained unchanged across groups. The gut virome, dominated by *Caudoviricetes*, showed high inter-individual diversity [75]. Further members of Proteobacteria order in HIV-1-infected individuals are increased, leading to a possibility of a decrease in Proteobacteria-infecting phages [59]. Thus, there is evidence and possibility of the vital role of bacteriome–virome interactions in maintaining gut homeostasis. However, there are challenges characterizing the gut virome: the unavailability of comprehensive data and viral sequences with their host specificity and interaction poses limitations that sometimes lead to biased and partial standards for the analysis of many gut viruses [76]. The literature thus far has only investigated DNA viruses, which limits applicability to varied forms of other viral genetic materials. Nevertheless, the attempt to understand the interplay of bacteriome, virome, and host factors will offer a unique opportunity to uncover disease-specific microbial patterns. Here, we discuss how bacteriome and virome, comprising eukaryotic viruses and phages, are altered in HIV-associated comorbidities related to metabolic disorders.

Moreover, metabolic syndrome includes many conditions like obesity, lipidemia, and insulin resistance. It was reported that the gut phage richness and diversity increased in obesity and obesity with metabolic syndrome compared to healthy individuals, suggesting disease-specific alterations [75]. Obese patients with T2DM have decreased gut viral richness and diversity as compared to lean and healthy controls. Namely, *Escherichia* phage, *Geobacillus* phage, and *Lactobacillus* phage were predominant in obese subjects, accompanied by weakened viral–bacterial correlations as compared to lean controls [77]. Compared to healthy controls, an abundance of *Caudoviricetes* phages over *Microviridae* is also reported in dysbiotic people. Alterations in the virome profile of individuals could be related to metabolic shifts, as seen when infection of murine norovirus–eukaryotic virus was carried out in antibiotic-treated mice. The infection protected the mice against antibiotic-associated intestinal injuries and bacterial infection [38]. One murine study provides evidence for a possible role of phages in activating gut mucosal immune responses, as in *Caudoviricetes*, which were shown to mediate TLR9-dependent activation of CD4 + T cells and increased intestinal inflammation by increasing the production of IFN-γ. While there are not very well-documented sources for gut virome-mediated immune regulation, the interaction could be predicted to be direct and indirect [78].

## 5. Common Microbiome Alterations in HIV-1 Infection and Type-2 Diabetes Mellitus (T2DM)

T2DM, a chronic metabolic disease, is a public health burden often attributed to causes such as nutrition transitions, urbanization, sedentary lifestyles, and stress. It is characterized by polydipsia, polyuria, weight loss, and polyphagia as the primary clinical manifestations. The key pathological factors are insulin resistance and impaired pancreatic β cell function. The gut microbiota is highly affected by diabetes and comorbidity of HIV-1 and T2DM. Butyrate producers such as *Finegoldia*, *Anaerococcus*, *Sneathia*, and *Adlercreutzia* are decreased, implying SCFA importance. A presumed reduction in anti-inflammatory functions and impaired insulin sensitivity is evident in dysbiosis. Prediabetic individuals show more abundant *Streptococcus* and *Anaerostignum* [79,80]. Both T2DM patients and PLWH tend to have reduced levels of *Akkermansia muciniphila*. Its depletion in PLWH is linked to worsened metabolic profiles and diminished CD8+ T cell function. *Fecalibacterium prausnitzii* is a major butyrate-producing bacterium, whose reduction in PLWH may worsen gut barrier dysfunction and systemic inflammation, which are both hallmarks of HIV-1 pathogenesis [81,82,83].

In the gut of T2D patients, more *Akkermansia muciniphila*, *Bacteroides intestinalis*, and *Escherichia coli* are found. Also, the reduction in *Fecalibacterium* and *Roseburia* observed in T2DM aligns with findings from HIV-1 infection. This implies the common microbial imbalances in both conditions that favor chronic inflammation and metabolic dysfunction [84].

One study found significant differences in the abundance of 58 phage species in fecal samples between T2DM patients and healthy controls. Phages specific to Enterobacteriaceae hosts, such as *Brochothrix*_phage_NF5, *Enterococcus*_phage_phiFL2A, *Streptococcus*_phage_PH10, and *Streptococcus*_phage_7201, showed significant variations between T2DM patients and non-diabetic individuals, thus suggesting T2DM may influence the composition of the gut phageome, potentially affecting gut health and microbial interactions [85].

In T2DM, members of the bacterial family Enterobacteriaceae are increased, along with the increase in bacteriophages belonging to the families *Siphoviridae*, *Podoviridae*, and *Myoviridae*, and unclassified members of the class *Caudoviricetes*, while *Flavobacterium* and *Cellulophaga* phage show a reduction in population [86].

*Lactobacillus* and *Prevotella* are elevated in both HIV-1 infection and T2DM, suggesting that these microbial groups play a role in the dysbiosis observed in these conditions. The abundance of *Prevotella* species, in particular, is possibly linked to insulin resistance; thus, similar phenomena are seen in comorbid conditions of HIV-1 and T2DM. *Bacteroides* species show decreased abundance in HIV-1 infection and an increase in T2DM, suggesting that the imbalance in *Bacteroides* may contribute significantly to dysbiosis in both conditions. Changes in gut virome in HIV-1 and T2DM comorbid individuals are seen as *Podoviridae* phages targeting *Escherichia* and *Clostridium*, and *Siphoviridae* phages infecting *Lactobacillus*, *Pseudomonas*, and *Staphylococcus* show an elevated abundance. These virome changes may contribute to altered microbial composition, reduced SCFA production, and chronic low-grade inflammation, all of which are linked to the progression of metabolic disorders like T2DM in HIV-1-positive individuals. The metabolism is also influenced by gut microbiota, as evidenced by an elevation in kynurenine/tryptophan ratio (KTR) in the tryptophan metabolism pathway. KTR indicates indoleamine-2,3-dioxygenase (IDO) activation, which facilitates the transformation of tryptophan into kynurenine, reducing tryptophan availability and altering gut and systemic immunity. Kynurenine shows bactericidal properties, but its accumulation also promotes regulatory T cell proliferation and apoptosis of T-effector cells, thus compromising host immunity in HIV-1 infection. In conditions such as T2DM, chronic inflammation driven by microbial dysbiosis leads to a similar elevation in KTR, further linking microbial changes to metabolic disturbances [87,88].

HIV-1 infection and T2DM exhibit distinct microbial changes, with some overlaps in their dysbiosis patterns. The comorbid state of HIV-1 and T2DM presents unique alterations, highlighting potential microbial contributions to disease progression and metabolic health.

## 6. Common Microbiome in HIV-1 Infection and Cardiovascular Diseases (CVD)

Obesity and high lipid content increase the chances of an individual developing cardiovascular diseases. Similar to diabetes, cardiovascular diseases are also a subset of conditions accompanied by gut microbiota changes. A marked depletion of beneficial bacteria from families like *Lachnospiraceae* and *Ruminococcaceae* produces SCFA-like butyrate. In contrast, pro-inflammatory taxa like *Gammaproteobacteria*, *Desulfovibrionaceae*, and *Ruminococcus gnavus* are seen. One study showed distinct gut dysbiosis in patients with obstructive coronary artery disease (CAD), including *Ruminococcus gnavus* and *Veillonella*. These are known producers of imidazole propionate (ImP), which shows higher levels in plasma in patients with obstructive CAD. Gut microbial metabolites, thus, are related to diseased conditions [89].

The reduced SCFA production and increased LPS are seen in the general population with CAD, which is common in the pathology of HIV-1. Studies of microbial metabolites such as ImP and their association with gut integrity, inflammation, and cardiovascular outcomes in PLWH could illuminate novel pathways linking gut dysbiosis to HIV-associated comorbidities, including heart failure, diabetes, and systemic metabolic disturbances. One study reported an increase of gut bacterial species, *Fusobacterium nucleatum*, to be associated with plaque formation, implying its role as a pathogenic factor in atherosclerosis development. Other gut bacterial species such as *Roseburia hominis*, *Roseburia inulinivorans*, *Johnsonella ignava*, *Odoribacter splanchnicus*, and *Clostridium saccharolyticum* are decreased with plaque formation in PLWH, highlighting their possible protective role [90]. A study on gut microbiota composition of abdominal obesity individuals showed a decrease in *Akkermansia*; thus, the factors that could lead to CVD are also associated with altered gut bacteriome. CVD patients showed increased *Siphoviridae*, *Myoviridae*, *Metaviridae*, and *Autographiviridae* while *Quimbyviridae* population is decreased [91]. However, limited evidence available is available regarding the role of viruses or bacteriophages in HIV-1 and CVD comorbidity [92].

HIV-1 infection and CVD exhibit distinct microbial changes, with some overlaps in their dysbiosis patterns. The comorbid state of HIV-1 and CVD presents unique alterations, highlighting potential microbial contributions to disease progression and metabolic health. The Table 1 summarizes the distinct and overlapping gut microbial shifts observed in individuals with HIV-1 infection, T2DM, and CVD, highlighting common taxa and dysbiosis patterns associated with their comorbid states.

## 7. Putative Crosstalk Between Bacteriome and Virome in HIV-Associated Metabolic Disorders

Considering crosstalk among various components of the microbiome, viz. the bacteriome, virome, and phageome, with the human host, is essential when assuming the whole consortia in the human gut, as they basically work as an ecosystem with all the components functioning in accordance. As discussed, HIV-1-induced alterations in the gut microbiota may interfere with normal host–microbiome communication. A study that followed Mendelian randomization and mediation analyses to examine the cause–effect relationship between HIV-1 and bacterial dysbiosis found that increased *Proteobacteria*, *Ruminococcaceae*, and Uncultured Clostridiales Group 013 (UCG013) are related to an increased risk of HIV-1 infection, while *Clostridium sensu stricto* and Erysipelotrichaceae were generally associated with less disease severity [88]. As a matter of fact, phages could regulate their bacterial host by following two well-known cycles—lytic or lysogenic. Additionally, some phages could exist as pseudo-lysogens, in which the phage genome exists in bacterial cells as a plasmid-like episomal construct without any integration or replication. For instance, *Plasmaviridae* could lead to a replication mode that can bud out without lysis and death of the bacterial host. Phages outnumber their bacterial hosts by a huge margin, as discussed previously. So, the remarkable adaptability of phages to change gut environments is understood in different diseased conditions. In PLWH, a depletion of beneficial bacteria (e.g., SCFA producers) suggests a possible increase in the abundance of phages infecting them. On the other hand, since pro-inflammatory species are predominant, the phages infecting them should decrease in the same setup. *Haemophilus* and *Sellimonas* supposedly mediate the regulation of specific metabolites, potentially influenced by phage–host interactions [93].

Another definitive example is *Bacteroides* phage B40-8 of the family *Siphoviridae* infecting *Bacteroides fragilis*—an important SCFA producer and immune modulator. This phage–bacterium interplay is likely affected in PLWH, presumably showing increased phage activity, further contributing to the depletion of the beneficial *Bacteroides* population. *Prevotella*, which is often increased in PLWH, produces LPS that may fuel systemic inflammation, worsening HIV-1 infection. Reduced populations of phages infecting *Prevotella* could signify an imbalance favoring pathogenic bacterial expansion [94].

As discussed earlier, the gut virome shifts a lot as an individual ages. Initially, the neonatal gut virome is colonized by prophages at birth, but the low bacterial abundance limits the lysogenic maintenance, so transiently, the lytic phages take over the abundance status. The virus-to-bacteria ratio rises as bacterial and viral densities increase. The maturation, aging, and diet changes lead to the colonization of a diverse community of prophage-harboring obligate anaerobes. So, the gut virome now comprises temperate phages that maintain overall lysogenic stability in the system, as seen in adult gut microbiomes [95]. By the age of 2, the *Microviridae* family of virulent phages is most prevalent, shifting from *Caudoviricetes* predominance at birth. p-crAssphage (lytic phages)-infecting Bacteroidetes become widespread among 2- to 5-year-old children. Lytic phage may infect dominant bacteria, causing cyclical shifts in bacterial abundance as seen in interactions between Bifidoprophages and *Bifidobacterium* in preterm infants [96,97].

The lytic phage phenotype often becomes more prominent during environmental stress and gut inflammation due to associated changes in the host’s microbiome and immune system. Although the whole mechanism is still under investigation, there is evidence to confirm that gut virome shifts from a lysogenic to a lytic state in diseased conditions [98]. Whether the transformation is a cause or effect of disease and how phages change during disease is still a topic of further research. Also noted here is the age-dependent lytic-to-lysogenic shift. In adulthood, the gut virome shift dominated by prophages offers advantages like metabolic benefits, antibiotic resistance, and increased bacterial competitiveness, as evident in *Vibrio cholerae* and its prophage, named CTXΦ [96].

Furthermore, there could be three applicable driving forces to understand the presumable shift from a lysogenic to a lytic state in diseased conditions.

Diversity-generating retroelements (DGRs): As evident in Lak phages with genomes exceeding 540 kilobases (kb), hence also called megaphages, the adaptability could be due to hypervariability caused by DGRs. DGRs could modify tail fiber proteins to alter host tropism, thus enabling one temperate phage—*Bacteroides dorei* Hankyphage—to infect a broad range of 13 *Bacteroides* species (Figure 3). *Bordetella* phage BPP-1 is another example of the same. Such DGR-containing phages are commonly integrated as prophages in bacterial hosts from groups Bacteroidetes, Firmicutes, and Proteobacteria. The lysogenic state predominating in the human gut is consistent with the “piggyback-the-winner” hypothesis, wherein phages benefit from maintaining prophage status during bacterial host replication. These prophages remain functional, producing viral particles and facilitating interactions within the microbial community, as made evident by viral metagenomics [99,100,101].

This schematic diagram shows the putative interplay between HIV-1 infection, gut bacteria, phages, and eukaryotic viruses, driving gut dysbiosis, chronic inflammation, and immune modulation. Leaky gut arises from epithelial barrier disruption, reduced tight junction proteins, and altered immune responses, leading to increased LPS production and inflammatory species expansion. A decline in SCFA producers worsens dysbiosis, while eukaryotic viruses like TTV modulate inflammation via CpG motifs, miRNAs, and NF-κB signaling. Phages influence microbial composition by infecting SCFA producers and inflammatory species. The lower panel highlights prophage-mediated bacterial regulation, where lysogeny and lytic shifts deplete beneficial bacteria, while stress-induced prophage activation leads to bacterial lysis through holins, endolysins, and lipases.

2.Dynamic genomic inversions (DGI): Recent research has shown DGI in bacteria such as *Bacteroides fragilis* illustrating phage–bacteria co-evolution. DGI could alter gene expression and affect host immunity, as seen in IBD. One study identified multiple invertible regions where a particular orientation was correlated with inflammatory bowel disease (IBD) as the promoter of polysaccharide A (PSA) of *B. fragilis* was mostly oriented “OFF” in IBD patients, correlating with the increased *B. fragilis*-associated bacteriophages. To further prove this, in mice colonized with a healthy human microbiota and *B. fragilis*, induction of colitis caused a decline of PSA in the “OFF” orientation. Thus, the lysogenic-to-lytic conversion in HIV and associated comorbidities related to metabolic diseases could be predicted due to DGI (Figure 3). Dynamic bacterial phase variations due to bacteriophages and host inflammation signify functional plasticity during diseased conditions [102].3.Holin/anti-holin secretion system: Balancing lysogeny and lysis is highly dependent on stressors and environmental cues. Gram-positive gut bacteria *Lactobacillus*, *Bifidobacterium*, and *Enterococcus* often harbor prophages. One mechanism for releasing these temperate phages to initiate the lytic cycle could be the holin/anti-holin system. Holins create pores in bacterial plasma membrane, enabling lysins to enter the peptidoglycan layer and lyse the host bacterial cell. But the timing is also of utmost importance, as the shift will be due to stresses like nutrient depletion, oxidative stress, or other dysbiosis caused by diseased conditions like HIV. To ensure the release of holins, anti-holin and anti-holin-like proteins are in place as gatekeepers releasing holins under specific conditions only. For Gram-negative bacteria such as *Prevotella* and *Bacteroides*, the lysis is a bit tricky due to an outer membrane (OM). Phages could use spanins to disrupt OM and lipases to degrade lipids, thus weakening the cell envelope. Once the OM is compromised, holins and endolysins complete the lytic cycle transition (Figure 3). The transition from temperate to lytic form in *Lactococcus lactis* MG1363 involves coordinated resonance between two prophages—TP712 (holin protein) and CAP—integrated into the bacterial genome. Mitomycin C treatment causes environmental stress during the transition. The CAP prophage induces lytic protein CAP endolysin, while TP712 creates pores, thus exemplifying the synergistic lysis and explaining the lytic phenotype of lysogens [103].

## 8. Conclusions

The gut microbiome is critical in maintaining metabolic, cardiovascular, and immune homeostasis. In people living with HIV, gut microbial dysbiosis is characterized by reduced diversity, depletion of beneficial SCFA-producing bacteria (e.g., *Lachnospiraceae* and *Ruminococcaceae*), and enrichment of pro-inflammatory taxa (e.g., *Ruminococcus gnavus* and *Gammaproteobacteria*). Gut virome analysis precludes the complexity and challenges, particularly in analyzing strain-specific interactions and estimating RNA viruses. Advancement in metagenomic sequencing and bioinformatics is required for detecting unknown viruses. We highlighted the intricate interplay between the human host, gut bacteriome, and virome comprising bacteriophages and eukaryotic viruses in healthy and diseased conditions of metabolic disorders associated with HIV-1 infection. The feed-forward cycle is critically discussed to observe the vicious, never-ending “feed” for prolonged inflammation, which is supposedly the commonality that joins these comorbidities. The regulatory mechanisms for adapting lytic or lysogenic cycles also depend on the host’s environment. The observed alterations in phage–host dynamics, bacterial composition, and virome diversity in HIV and comorbidities imply the essentiality of considering the virome’s potential as a modulator of disease. Future research should focus on unraveling the precise mechanisms through which gut viruses contribute to disease pathogenesis and exploring phage-driven precision therapies for restoring gut homeostasis in HIV-1 and associated metabolic disorders.

## 9. Challenges and Future Perspectives

Existing studies on alterations in the human gut microbiome across various diseases suggest a strong link between gut microbiome dysbiosis, including virome imbalance, and disease pathophysiology. However, the field is still emerging, and multiple key questions remain unanswered. Few studies explore the causal relationship between gut microbial communities and disease, limiting our understanding of how these interactions may promote or inhibit disease onset and progression. Specifically, the correlation versus causation dynamic between gut bacteriome and phageome dysbiosis in different diseases remains poorly defined. Future research must shift from correlative observations to mechanistic investigations to unravel the causal relationships between alterations in gut microbial communities, particularly the bacteriome and phageome, and disease onset, progression, or resolution. Virome research faces significant challenges due to limited computational tools and incomplete reference databases, which are biased toward known viruses, predominantly DNA viruses. Consequently, the gut virome remains underexplored, with more than 50% of viral sequences identified by metagenomic sequencing remaining unclassified. This gap hinders the accuracy and reliability of generalizations drawn from current studies. The RNA virome has received less attention than the DNA virome. The integration of longitudinal multi-omics approaches, improved viral annotation techniques, and robust in vitro and in vivo models will be pivotal in advancing virome research. These developments hold promise not only for enhancing diagnostic precision but also for guiding the design of next-generation microbiome-based interventions. In particular, fecal virome transplants and targeted phage therapies may emerge as refined therapeutic strategies to restore microbial balance and treat disease with higher specificity. Moreover, the potential of the fecal virome and phageome as non-invasive biomarkers for disease diagnostics and prognostics deserves deeper exploration. As our understanding of the virome matures, it may serve as a critical component in the personalized medicine landscape, offering new avenues for early detection and therapeutic modulation of disease.

## Figures and Tables

**Figure 1 viruses-17-00990-f001:**
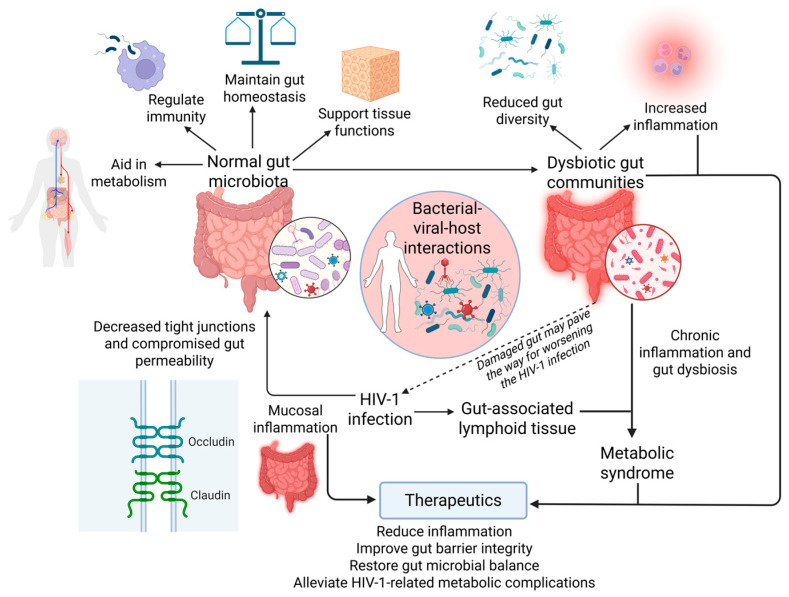
Gut microbiota–virome–HIV-1 axis: mechanisms and therapeutic insights. Created in https://BioRender.com, accessed on 4 July 2025.

**Figure 2 viruses-17-00990-f002:**
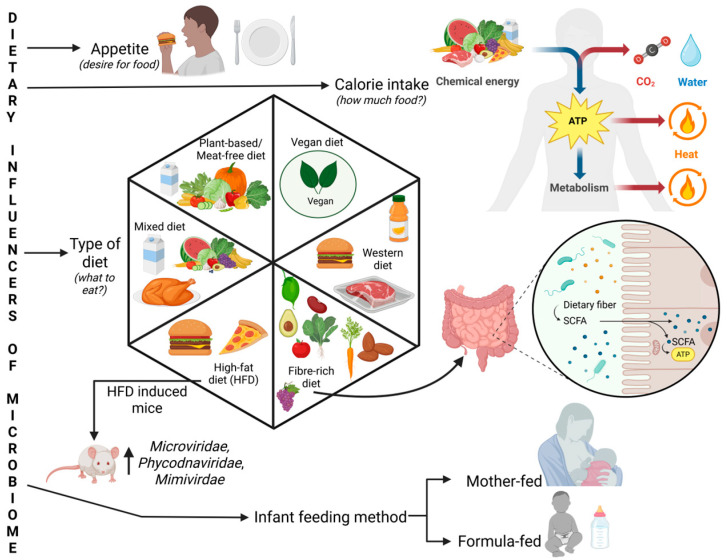
Dietary influencers of the gut microbiome with their metabolic implications. Created in https://BioRender.com, accessed on 4 July 2025.

**Figure 3 viruses-17-00990-f003:**
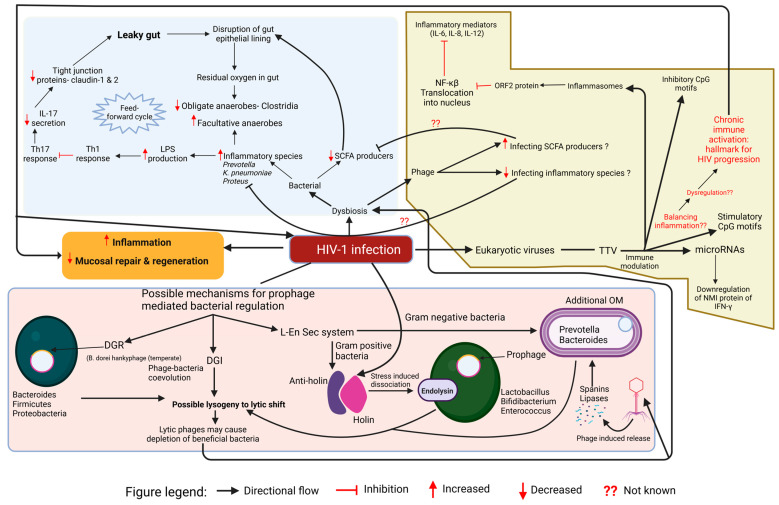
Putative Crosstalk between HIV-1 Infection, Gut Microbiome, and Virome in HIV Disease progression. Created in https://BioRender.com, accessed on 4 July 2025.

**Table 1 viruses-17-00990-t001:** Comparative gut microbial alterations in HIV-1 infection and related comorbidities.

Population	Increased Bacterial Species	Increased Viral Population	Decreased Bacterial Species	Decreased Viral Population
HIV-1	*Enterobacteriaceae*	Adenovirus	*Rikenellaceae*	*Myoviridae*
	*Desulfovibrio (Proteobacteria)*		*Bacteroides*	Podoviridae
	Erysipelotrichaceae		Bacteroidaceae (Bacteroidetes)	
	Veillonellaceae		Alistipes	
	*Dialister*		Fecalibacterium	
	*Mitsuokella*		Barnesiella	
	*Catenibacterium*		Lachnospira	
	*Mogibacterium*		Ruminococcaceae	
	*Bulleidia*			
	*Lactobacillus*			
	*Prevotella*			
T2D	*Streptococcus*	Phages specific to *Enterobacteriaceae*	*Finegoldia*	*Flavobacterium phage*
	*Akkermansia muciniphila*	*Siphoviridae*	*Akkermansia* spp.	*Cellulophaga phage*
	*Bacteroides intestinalis*	*Podoviridae*	*Christensenellaceae R7 group*	
	*Escherichia coli*	*Myoviridae*	*Anaerococcus*	
	*Anaerostignum*	Unclassified members of *Caudoviricetes*	*Sneathia*	
			*Adlercreutzia*	
CVD	Gammaproteobacteria	*Siphoviridae*	*Johnsonella ignava*	*Quimbyviridae*
	*Veillonella*	*Myoviridae*	*Odoribacter splanchnicus*	
	*Fusobacterium nucleatum*	*Metaviridae*	*Clostridium accharolyticum*	
		*Autographiviridae*		
HIV-1 + T2DM	*Prevotella*	*	*Fecalibacterium*	*
			*Akkermansia muciniphila*	
			*Roseburia*	
			*Bacillota*	
			*Ruminococcus*	
HIV-1 +CVD	*Desulfovibrio (Proteobacteria)*	*	*Lachnospira*	*
			*Roseburia hominis*	
			*Ruminococcaceae*	
			*Roseburia inulinivorans*	

* Population not found as of 30 June 2025 in PubMed or Google Scholar search engines.

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
