# Peer review of "Possible Crosstalk and Alterations in Gut Bacteriome and Virome in HIV-1 Infection and the Associated Comorbidities Related to Metabolic Disorder"

_viruses, 2025, doi:10.3390/v17070990_

Round 1
Reviewer 1 Report
Comments and Suggestions for Authors
The authors of this manuscript present a comprehensive review on the alterations of the gut microbiome and virome in HIV-1-infected individuals, with a particular focus on the interplay between microbial dysbiosis, immune dysfunction, and metabolic comorbidities such as type 2 diabetes and cardiovascular disease. The topic addressed is timely, relevant, and targets a specific and clinically important patient population. Below are my suggestions for improving the quality of the manuscript.
Major Revisions
1. Abstract
- Lines 13–14: The sentence “Increased life expectancy and better medication is allowing People living with HIV (PLWH) to live more” should be revised. A more appropriate phrasing would be:
“Improved antiretroviral therapy (ART) has significantly increased the life expectancy of people living with HIV (PLWH).”
- The abstract would benefit from improved cohesion and structure. For example, lines 22–23 and line 33 mention similar terms (e.g., phage therapy, faecal virome transplantation). A clearer and more logical flow could be achieved by starting with the general context, proceeding to the aim of the review, and concluding with the scientific contributions and implications of the study.
2. Introduction
- The concept of dysbiosis should be more clearly defined early in the introduction, though without extensive elaboration.
- The introduction is somewhat lengthy. I suggest condensing it to improve readability and ensure a smooth transition to the main sections of the review.
- A commonly used narrative review structure would be to summarize the essential background concisely and conclude the introduction with a short paragraph explicitly stating the article’s objective.
3. Line 405: The statement that patients with T2DM have a gut microbiome “rich in Akkermansia muciniphila” is inaccurate. On the contrary, both T2DM patients and PLWH tend to have reduced levels of Akkermansia muciniphila. This should be corrected.
4. The manuscript would be strengthened by including more discussion on beneficial bacteria such as Akkermansia muciniphila and Faecalibacterium prausnitzii, which are commonly reduced in PLWH. The following recent references may prove useful:
- Guo Y, Tang G, Wang Z, Chu Q, Zhang X, Xu X, Fan Y. Characterization of the gut microbiota in different immunological responses among PLWH. Sci Rep. 2025 Apr 24;15(1):14311. doi: 10.1038/s41598-025-98379-0.
- Diesse JM, Jadhav S, Tamekou SL, Simo G, Dzoyem JP, Souopgui J, Kuiate JR, Nema V. Disturbances in the gut microbiota potentially associated with metabolic syndrome among patients living with HIV-1 and on antiretroviral therapy at Bafoussam Regional Hospital, Cameroon. Diabetol Metab Syndr. 2025 Mar 15;17(1):86. doi: 10.1186/s13098-025-01653-4.
- Qian Z, Chen S, Liao X, Xie J, Xu Y, Zhong H, Ou L, Zuo X, Xu X, Peng J, Wu J, Cai S. Decreased intestinal abundance of Akkermansia muciniphila is associated with metabolic disorders among people living with HIV. Ann Med. 2025 Dec;57(1):2474730. doi: 10.1080/07853890.2025.2474730.
Minor Revisions
1. In the title, the word co-morbidity should be corrected to comorbidity.
2. Ensure consistency throughout the manuscript with abbreviations. For example, an abbreviation introduced in line 50 should not appear for the first time in line 51.
3. Consistent terminology should be used—e.g., the cytokine interferon-gamma should be written as interferon-gamma (IFN-γ) (line 94).
4. Use italics for Latin species names where required, such as Fusobacterium nucleatum (line 469).
5. In the keywords, “HIV-1” could be removed, as the full term is already included.
6. Section 7 may be better placed after the Introduction to improve logical flow.
7. Section 9 could be more appropriately titled Challenges and Future Perspectives to reflect its content more clearly.
Author Response
We thank the reviewer for investing time and expertise in suggesting possible ways of improving our manuscript to reach the level of acceptance in the esteemed journal Viruses. We have tried our best to make good use of their contribution and modified the manuscript accordingly.
- Abstract
Query: Lines 13–14: The sentence “Increased life expectancy and better medication is allowing People living with HIV (PLWH) to live more” should be revised. A more appropriate phrasing would be:“Improved antiretroviral therapy (ART) has significantly increased the life expectancy of people living with HIV (PLWH).”
Response: As per the suggestion of reviewer, we have revised the sentence and incorporated it in line 14-15
Query: The abstract would benefit from improved cohesion and structure. For example, lines 22–23 and line 33 mention similar terms (e.g., phage therapy, faecal virome transplantation). A clearer and more logical flow could be achieved by starting with the general context, proceeding to the aim of the review, and concluding with the scientific contributions and implications of the study.
Response: We have revised the abstract to improve logical flow and cohesion by beginning with the broader context, followed by the key findings, and concluding with potential therapeutic implications. Redundancies (e.g., repeated mention of phage therapy and FVT) have been removed or consolidated to enhance clarity. The revised abstract now presents a clearer progression from background to aim and implications.
- Introduction Query: The concept of dysbiosis should be more clearly defined early in the introduction, though without extensive elaboration. Response: Dysbiosis in the simplest form as disruption is introduced in line 39-40.
Query: The introduction is somewhat lengthy. I suggest condensing it to improve readability and ensure a smooth transition to the main sections of the review.
Response: We have shortened the introduction and made it simpler with crisp information about the content and the concept.
Query: A commonly used narrative review structure would be to summarize the essential background concisely and conclude the introduction with a short paragraph explicitly stating the article’s objective.
Response: We have revised the end of the introduction to include a clear and concise statement of the article’s objective. Specifically, we now highlight our hypothesis that shifts in gut virome composition contribute to chronic inflammation in HIV-1-associated metabolic comorbidities through phage–host immune interactions.
Line 405: The statement that patients with T2DM have a gut microbiome “rich in Akkermansia muciniphila” is inaccurate. On the contrary, both T2DM patients and PLWH tend to have reduced levels of Akkermansia muciniphila. This should be corrected. The manuscript would be strengthened by including more discussion on beneficial bacteria such as Akkermansia muciniphila and Faecalibacterium prausnitzii, which are commonly reduced in PLWH. The following recent references may prove useful:
- Guo Y, Tang G, Wang Z, Chu Q, Zhang X, Xu X, Fan Y. Characterization of the gut ...
- Diesse JM, Jadhav S, Tamekou SL, Simo G, Dzoyem JP, Souopgui J, Kuiate JR, Nema V. Disturbances in the gut …
- Qian Z, Chen S, Liao X, Xie J, Xu Y, Zhong H, Ou L, Zuo X, Xu X, Peng J, Wu J, Cai S. Decreased intestinal abundance.... Response: We acknowledge the inaccuracy and have corrected the statement accordingly in line 414-419 with inclusion of suggested 3 references. The revised text now reflects that Akkermansia muciniphila levels are reduced in both T2DM patients and PLWH, and highlights its association with worsened metabolic outcomes and impaired CD8⁺ T-cell function in PLWH. We also added context regarding the depletion of Faecalibacterium prausnitzii and its relevance to gut barrier integrity and inflammation with the suggested references by the reviewer.
Minor Revisions
1. In the title, the word co-morbidity should be corrected to comorbidity.
Response: Addressed (line no. 3)
- Ensure consistency throughout the manuscript with abbreviations. For example, an abbreviation introduced in line 50 should not appear for the first time in line 51. Response: Addressed (line 49-50)
- Consistent terminology should be used—e.g., the cytokine interferon-gamma should be written as interferon-gamma (IFN-γ) (line 94). Response: Addressed as “interferon-gamma (IFN- γ)]” in line no. 89
- Use italics for Latin species names where required, such as Fusobacterium nucleatum(line 469). Response: Addressed in line 478. Further in whole manuscript, Species, Genus, and families are italicized. As per ICTV (International Committee on Taxonomy of Viruses), viral family names are also italicized.
- In the keywords, “HIV-1” could be removed, as the full term is already included. Response: Addressed in line 33-34
- Section 7 may be better placed after the Introduction to improve logical flow. Response: Replaced as suggested
- Section 9 could be more appropriately titled Challenges and Future Perspectives to reflect its content more clearly. Response: Incorporated the change as suggested
Reviewer 2 Report
Comments and Suggestions for Authors
The manuscript submitted to Viruses by Vijay Nema et al. is devoted to changes in the composition of viruses and bacteria in the gut during HIV and metabolic syndrome.
The reviewer believes that the topic of the article is unlikely to be of interest to a wide range of readers, however, this topic deserves a review. The list of references contains almost 100 references, many of them published in the last 5-10 years
Review articles are much better cited when they contain tables, figures etc. These should present mechanisms of actions, physiological effects, etc and compare data from several sources. In this paper, the authors made two very similar figures (Fig 1 and 2), the value of which as a figure is low and, perhaps, the presentation of this material as tables would be even more useful for the reader. There is also presented Fig 3, which corresponds to the recommendations that the reviewer would like to give to the authors.
Thus, I suggest that the authors add to all sections tables and/or figures, which will undoubtedly decorate the article and make it more interesting and useful for the reader.
Author Response
We thank the reviewer for investing time and expertise in suggesting possible ways of improving our manuscript so that it would be accepted in the esteemed journal Viruses. We have tried our best to make good use of their contribution and modified the manuscript accordingly.
Query:
The manuscript submitted to Viruses by Vijay Nema et al. is devoted to changes in the composition of viruses and bacteria in the gut during HIV and metabolic syndrome.
Response:
Thanks for summarizing our work
Query:
The reviewer believes that the topic of the article is unlikely to be of interest to a wide range of readers, however, this topic deserves a review. The list of references contains almost 100 references, many of them published in the last 5-10 years
Response:
Now we have 103 references after addition of suggested papers.
Query:
Review articles are much better cited when they contain tables, figures etc. These should present mechanisms of actions, physiological effects, etc and compare data from several sources.
In this paper, the authors made two very similar figures (Fig 1 and 2), the value of which as a figure is low and, perhaps, the presentation of this material as tables would be even more useful for the reader. There is also presented Fig 3, which corresponds to the recommendations that the reviewer would like to give to the authors.
Thus, I suggest that the authors add to all sections tables and/or figures, which will undoubtedly decorate the article and make it more interesting and useful for the reader.
Response:
Thank you for this insightful suggestion. In response, we have replaced the two Venn diagram figures (Fig. 1 and Fig. 2) with a consolidated table that more effectively compares gut microbial alterations across HIV-1 infection, CVD, and T2DM, supported by references cited in the text. Additionally, we have included a new figure summarizing dietary influences on the gut microbiome, further enhancing the mechanistic and physiological context as desired. We appreciate your recommendation, which has helped us strengthen the visual and comparative content of the manuscript.
Round 2
Reviewer 2 Report
Comments and Suggestions for Authors
The manuscript has been reworked by the authors and can be recommended for publication